# Cisplatin Plus Cetuximab Inhibits Cisplatin-Resistant Human Oral Squamous Cell Carcinoma Cell Migration and Proliferation but Does Not Enhance Apoptosis

**DOI:** 10.3390/ijms22158167

**Published:** 2021-07-29

**Authors:** Hyeong Sim Choi, Young-Kyun Kim, Pil-Young Yun

**Affiliations:** 1Department of Oral and Maxillofacial Surgery, Section of Dentistry, Seoul National University Bundang Hospital, 82 Gumi-ro 173 beon-gil, Bundang-gu, Seongnam 13620, Korea; r2013@snubh.org (H.S.C.); kyk0505@snubh.org (Y.-K.K.); 2Department of Dentistry and Dental Research Institute, School of Dentistry, Seoul National University, 101 Daehak-ro, Jongno-gu, Seoul 03080, Korea

**Keywords:** oral squamous cell carcinoma, cisplatin, cetuximab, epithelial-to-mesenchymal transition, migration, epidermal growth factor receptors

## Abstract

Cisplatin is among the most widely used anticancer drugs used in the treatment of several malignancies, including oral cancer. However, cisplatin treatment often promotes chemical resistance, subsequently causing treatment failure. Several studies have shown that epidermal growth factor receptors (EGFRs) play a variety of roles in cancer progression and overcoming cisplatin resistance. Therefore, this study focused on EGFR inhibitors used in novel targeted therapies as a method to overcome this resistance. We herein aimed to determine whether the combined effects of cisplatin and cetuximab could enhance cisplatin sensitivity by inhibiting the epithelial-to-mesenchymal transition (EMT) process in cisplatin-resistant cells. In vitro analyses of three cisplatin-resistant oral squamous cell carcinoma cells, which included cell proliferation assay, combination index calculation, cell cytotoxicity assay, live/dead cell count assay, Western blot assay, propidium iodide staining assay, scratch assay, and qRT-PCR assay were then conducted. Our results showed that a cisplatin/cetuximab combination treatment inhibited cell proliferation, cell motility, and N-cadherin protein expression but induced E-cadherin and claudin-1 protein expression. Although the combination of cisplatin and cetuximab did not induce apoptosis of cisplatin-resistant cells, it may be useful in treating oral cancer patients with cisplatin resistance given that it controls cell motility and EMT-related proteins.

## 1. Introduction

Studies have shown that the overexpression of the epidermal growth factor receptor (EGFR), a member of the ErbB family of receptors, was associated with the development of various cancers [1,2,3]. Moreover, evidence has found that the activation of the EGFR signaling pathway promotes the proliferation, invasion, and metastasis of cancer cells [4,5,6]. The diverse roles of the EGFR in cancer progression have facilitated the development of anticancer therapies that interfere with EGFR-mediated effects, such as monoclonal antibodies and small-molecule tyrosine kinase inhibitors [4,7,8].

Cetuximab (trade name Erbitux) is an anti-EGFR monoclonal antibody administered through intravenous infusion used to treat metastatic colorectal cancer and head and neck cancer [7,8,9]. Recent *in vitro* studies have reported that: a specific EGFR tyrosine kinase inhibitor increased the therapeutic effects of cisplatin in oral squamous cell carcinoma (OSCC) cells; cetuximab increased cisplatin-induced apoptosis through the inactivation of the EGFR/AKT signaling pathway in nasopharyngeal carcinoma (NPC); cetuximab plus platinum-based chemotherapy enhanced the overall survival in patients with recurrent or metastatic squamous cell carcinoma of the head and neck; and cetuximab improved cisplatin-induced reticulum stress-related apoptosis in laryngeal squamous cell carcinoma cells by suppressing the expression of TXNDC5 [9,10,11]. Moreover, other reports have shown that cetuximab attenuated cell invasion/metastasis-related processes in gastric cancer, whereas E-cadherin protein expression correlated with cetuximab sensitivity in non-small cell lung cancer (NSCLC) [12,13].

Cadherins are a class of intercellular adhesion molecules important for the formation of adherens junctions that bind cells with each other and are essential for maintaining cell-cell contact and regulating cell-cell adhesion among different cells [14,15]. E- and N-cadherin belong to type-I classical cadherins [15]. E-cadherin plays an important role in tumor suppression considering that the downregulation of E-cadherin expression or function facilitated increased invasion in malignant epithelial cancers [15,16]. On the other hand, N-cadherin expression in cancer cells enhances cancer cell motility and promotes cancer metastasis [14,15].

While epithelial-to-mesenchymal transition (EMT) is a developmental process generally observed in normal embryogenesis, this process can also occur during wound healing and the initiation of metastasis during cancer progression [16,17,18]. Throughout the process of EMT, the decreased expression of epithelial markers (e.g., E-cadherin and claudin) and the increased expression of mesenchymal markers (e.g., N-cadherin and vimentin) have been reported [16,19]. Recent studies have increasingly reported that EMT not only contributes to the metastasis of cancer cells but also plays an important role in anticancer drug resistance after chemotherapy treatment [20,21,22,23]. Thus, inhibition of this cellular process may constitute a method for overcoming chemoresistance [20,21,24].

Our established cisplatin-resistant human OSCC cell lines showed greater N-cadherin protein expression and cell motility compared to their parental cell lines [25]. Therefore, the current study attempted to determine whether the combined treatment of cisplatin and cetuximab affected apoptosis and cell motility. The present study demonstrated the in vitro activity of cetuximab in three cisplatin-resistant OSCC cell lines and showed that an EGFR blockade with cisplatin decreased the proliferation and migration thereof.

## 2. Results

### 2.1. Cisplatin and Cetuximab Co-Treatment Promoted Greater Inhibition of Cisplatin-Resistant OSCC Cell Proliferation Compared to Each Alone

To study the effects of combining cisplatin and cetuximab on OSCC cell growth, an MTT assay was performed. First, three cisplatin-resistant cell lines (YD-8/CIS, YD-9/CIS, and YD-38/CIS) were treated with each drug at various concentrations for 72 h. Accordingly, cisplatin treatment dose-dependently inhibited cell growth in YD-9/CIS and YD-38/CIS (Figure 1a), although only a slight suppression was observed in YD-8/CIS. However, cetuximab had no effect on the growth of all three cell lines (Figure 1b). Thereafter, we determined whether co-treatment with cisplatin (1–5 μg/mL) and cetuximab (200 or 500 μg/mL) had a synergistic effect on cell growth (data not shown). Combination treatment with both drugs at a low dose (1 μg/mL of cisplatin and 200 μg/mL of cetuximab) promoted greater cell growth inhibition compared to each treatment administered alone (Figure 1c). Moreover, combination index (CI) analysis found synergism in most combinations, except for a few cases in YD-8/CIS cells (Figure 1d–i). Moreover, cetuximab tended to have a greater synergistic effect in combination with cisplatin at 200 μg/mL than at 500 μg/mL. The aforementioned data showed a synergistic effect between cisplatin (1 μg/mL) and cetuximab (200 μg/mL).

### 2.2. Cisplatin and Cetuximab Co-Treatment Inhibited Cisplatin-Resistant OSCC Cell Growth but Did Not Induce Cell Death

To confirm whether the synergistic effects were due to cytotoxicity, we measured the amount of lactate dehydrogenase (LDH) enzyme released into the surrounding cell culture medium. Upon exposure to toxic compounds and plasma membrane damage, cells release LDH into the cell culture medium. Therefore, the three cisplatin-resistant cell lines were treated with cisplatin (1 μg/mL) and/or cetuximab (200 μg/mL) for 15 h. Given that the animal serum used in our experiment contained varying amounts of LDH, the cell culture medium containing 1% serum was used in this assay. Figure 2a shows minimal changes in cell cytotoxicity in each cell.

We subsequently performed a trypan blue assay to determine whether cisplatin (1 μg/mL) and cetuximab (200 μg/mL) co-treatment could induce cell death. This method is based on the principle that living (viable) cells do not uptake the trypan blue dye, whereas dead (non-viable) cells absorb it. As such, only the number of living cells was measured in Figure 2b, whereas the number of both dead and living cells was measured in Figure 2c–e. Figure 2b–e show that cisplatin and cetuximab co-treatment promoted a greater reduction in the number of living cells compared to their individual treatment, whereas the number of dead cells tended to be approximately the same or slightly increased.

To characterize EGFR signaling, which may be related to the synergistic inhibitory effect of cisplatin and cetuximab on cisplatin-resistant OSCC cells, the levels of EGFR and its phosphorylation signaling pathway proteins were determined through Western blotting. Figure 2f shows that cetuximab promoted little change in EGFR protein expression in all three cell lines but significantly reduced the expression of two p-EGFR (Y992 and Y1068) proteins in YD-38/CIS. However, co-treatment with cisplatin and cetuximab markedly decreased the expression of p-EGFR (Y1045) protein in YD-8/CIS and YD-38/CIS.

Therefore, the aforementioned results suggested that co-treatment with cisplatin and cetuximab induced growth inhibition by limiting the number of viable cells rather than killing them.

### 2.3. Cisplatin and Cetuximab Co-Treatment Did Not Induce Apoptosis in Cisplatin-Resistant OSCC Cells

To evaluate whether cisplatin (1 μg/mL) and cetuximab (200 μg/mL) co-treatment inhibited cell growth by enhancing apoptosis, propidium iodide (PI) staining, and Western blot assays were performed. In all cisplatin-resistant OSCC cells, cisplatin and cetuximab individually and in combination, promoted little change in the sub-G_1_ phase population (Figure 3a). Similarly, the cisplatin and cetuximab combination slightly increased the expression of caspase-9, -7, and -3 and PARP proteins (Figure 3b). Overall, our results showed that cetuximab did not enhance cisplatin-mediated apoptosis in cisplatin-resistant OSCC cells.

### 2.4. Cisplatin and Cetuximab Combination Restored Epithelial Cell Properties with Increased Expression of Adhesion Molecules but Decreased Cell Motility and Mesenchymal Maker Expression in Cisplatin-Resistant OSCC Cells

Our previous study reported that three cisplatin-resistant OSCC cells acquired the EMT phenotype. Thus, we determine whether the combination of cisplatin (1 μg/mL) and cetuximab (200 μg/mL) in three cisplatin-resistant OSCC cell lines was associated with EMT characteristics through in vitro scratch, Western blot, and qRT-PCR assays. Figure 4a shows that the combination of cisplatin and cetuximab significantly reduced the number of cells migrating into the wounded area in cisplatin-resistant OSCC cells. Moreover, combination therapy promoted greater E-cadherin protein expression but lower N-cadherin protein expression in the cisplatin-resistant cells compared to other treatments (Figure 4b). Although claudin-1 protein expression was increased in YD-8/CIS and YD-9/CIS, no change was observed in YD-9/CIS. Furthermore, vimentin protein expression was decreased only in YD-9/CIS. However, the combination of cisplatin and cetuximab had little effect on the expression of *E-cadherin* and *N-cadherin* (Figure 4c). The aforementioned findings indicate that the combination therapy caused molecular changes consistent with cell migration and EMT signaling pathway proteins.

## 3. Discussion

Cisplatin, a DNA-damaging agent, is a well-known anticancer drug widely used in the chemotherapy of various human cancers, including oral cancers [9,26,27]. Nevertheless, patients receiving cisplatin-based treatments often exhibit lower response rates during subsequent treatment [28,29]. This problem can be attributed to several reasons, with some studies suggesting the importance of EMT in the development of resistance to cisplatin treatment [28,29,30,31]. Cisplatin-resistant OSCC cells established by our group were characterized by enhanced EMT markers [25]. Moreover, other studies have shown that EGFR could overcome cisplatin resistance by restoring cisplatin sensitivity through an EGFR blockade in cisplatin-resistant epithelial ovarian cancer cells and OSCC cells [1,5]. Thus, the current study aimed to determine whether the combined effects of cisplatin and cetuximab could enhance cisplatin sensitivity by inhibiting the EMT process in cisplatin-resistant cells.

Accordingly, our results showed that cetuximab and cisplatin synergistically reduced the proliferation of cisplatin-resistant OSCC cells, with their synergistic effects being confirmed using CI analysis. Moreover, the present study found that cisplatin/cetuximab co-treatment did not induce apoptosis but instead inhibited the increase in cell number. After subsequently evaluating whether the synergistic effects of cisplatin and cetuximab on cell growth inhibition was related to EMT, we observed that cisplatin/cetuximab co-treatment inhibited cell motility and regulated EMT markers, such that E-cadherin and claudin-1 proteins were upregulated, whereas N-cadherin was downregulated.

The current study revealed that the synergistic effects of cetuximab and low dose cisplatin co-treatment potentially suppressed cell migration and the expression levels of EMT markers, a hallmark of increased cisplatin-resistant cells. However, given the yet unclear molecular mechanisms for the synergistic effects of cisplatin and cetuximab, further studies are warranted. In conclusion, the combination of cisplatin and cetuximab may be a useful treatment strategy for patients with oral cancer who have acquired cisplatin resistance.

## 4. Materials and Methods

### 4.1. Reagent

Cisplatin (PubChem CID: 84691) (PubChem, Bethesda, MD, USA), dissolved in distilled water at 1 mg/mL, was purchased from JW Medical C. (JW Medical C., Seoul, Korea). Cetuximab, a monoclonal antibody administered via intravenous infusion and distributed under the trade name Erbitux, was obtained from Merck KGaA (Merck KGaA, Darmstadt, Germany).

### 4.2. Cell Lines and Cell Cultures

Cisplatin-resistant OSCC cell lines (YD-8/CIS, YD-9/CIS, and YD-38/CIS) were derived from their parental OSCC cell lines (YD-8, YD-9, and YD-38) using methods described in our previous paper [25].

### 4.3. Cell Proliferation Assay

Cells (5 × 10^3^ cells/well in 50 μL) were seeded into 96-well plates and incubated in the presence of various amounts of cisplatin and cetuximab for 72 h. MTT reagent (0.5 mg/mL) was added into each well, after which cells were reincubated for another 2 h. Thereafter, the medium was discarded, and the remaining formazan formed were dissolved in 100 μL of DMSO and quantitated at 570 nm using a SpectraMax Plus 384 microplate reader (Molecular Devices, LLC., San Jose, CA, USA).

### 4.4. Combination Index Calculation

The synergistic effects of cisplatin and cetuximab were automatically simulated using CompuSyn 1.0 software (ComboSyn, Inc., Paramus, NJ, USA). Values were defined according to previous studies. Accordingly, CI < 0.9 indicated synergism; 0.9 ≤ CI ≤ 1.1 indicated additive effects; and CI > 1.1 indicated antagonism [32,33,34].

### 4.5. Cell Cytotoxicity Assay

Lactate dehydrogenase (LDH) is a cytosolic enzyme and a well-established indicator of cellular toxicity. During cell damage, LDH is released from the plasma membrane into the surrounding cell culture medium. Briefly, cells (2.5 × 10^4^ cells/well in 200 μL) were seeded into 48-well plates and then treated with cisplatin (1 μg/mL) and cetuximab (200 μg/mL) in the presence of 1% serum. After incubation for 15 h, the plate was centrifuged at 250× *g* for 10 min. Thereafter, 100 μL of supernatant from each well was carefully transferred into a clear 96-well plate, followed by the addition of 100 μL of the reaction mixture from the CytoTox 96® Non-Radioactive Cytotoxicity Assay (Promega, Madison, WI, USA) and incubation for 30 min at room temperature while blocking the light. The absorbance of the samples was measured at 490 nm using a SpectraMax Plus 384 microplate reader (Molecular Devices, LLC., San Jose, CA, USA).

### 4.6. Live/Dead Cell Count Assay

For the live/dead cell count assay, cells in suspension were stained with trypan blue, which is used to count live cells by labeling only dead cells. First, cells (2.5 × 10^4^ cells/well in 200 μL) were seeded into 48-well plates and then incubated in the presence of cisplatin (1 μg/mL) and cetuximab (200 μg/mL) for 72 h. Cells were then collected into a conical tube, stained with trypan blue, and pipetted into a disposable Countess^®^ chamber slide (Thermo Fisher Scientific, Waltham, MA, USA). The slide was inserted into the Countess™ II automated cell counter (Thermo Fisher Scientific, Waltham, MA, USA), after which the number of cells was counted.

### 4.7. Western Blot Assay

To investigate the expression levels of EGFR proteins (EGFR and phosphor-EGFR (p-EGFR; Y992, Y1045, and Y1068)), apoptosis signaling pathway proteins (Caspase-9, -7, -3, and PARP), EMT signaling pathway proteins (E-cadherin, Claudin-1, Vimentin, N-cadherin), and the housekeeping protein (GAPDH) in the cell lines, electrophoresis, and blotting were performed. All antibodies were purchased from Cell Signaling Technology (Cell Signaling Technology, Inc., Danvers, MA, USA). Immunobands were detected using the EZ-western detection kit (Daeil Lab Service Co., Ltd, Seoul, Korea) and visualized using an automatic X-ray film processor JP-33 (JPI Healthcare Co., Ltd., Seoul, Korea).

### 4.8. Propidium Iodide Staining Assay

DNA content distribution was quantified using the PI/RNase Staining Buffer (BD, Franklin Lakes, NJ, USA). Each cell was fixed with 70% ethanol, stained with PI, and then observed, with apoptotic cells being represented by the sub-G_1_ population. The fluorescence intensity for each cell was detected via flow cytometry using a FACSCalibur instrument (BD, Franklin Lakes, NJ, USA), and data were analyzed using BD CellQuest Pro 6.0 software (BD, Franklin Lakes, NJ, USA).

### 4.9. In Vitro Scratch Assay

To measure cell migration *in vitro*, cells (2.5 × 10^4^ cells/well in 200 μL) were seeded into a 48-well plate. Once the cells reached 100% confluence, the cell monolayer was scratched with a sterile pipette tip. After incubation for 18–19 h, the area with remaining cells was washed with PBS. The cells that migrated into the scratch region were imaged using an Olympus CKX53 inverted microscope (Olympus Corporation, Shinjuku, Tokyo, Japan).

### 4.10. RNA Isolation and Real-Time qPCR

RNA was extracted from collected cells using an RNA-spin™ Total RNA Extraction Kit (iNtRON Biotechnology, Inc., Seongnam, Korea). cDNA was directly synthesized from the RNA using High-Capacity cDNA Reverse Transcription Kits (Thermo Fisher Scientific, Waltham, MA, USA). The expression levels of *E-cadherin*, *N-cadherin*, and *GAPDH* were analyzed using a qPCR QuantStudio 7 Flex Real-Time PCR System using GoTaq^®^ qPCR Master Mix (Promega, Madison, WI, USA). The following primers were used: *E-cadherin*, 5′-CAGAAAGTTTTCCACCAAAG-3′ (forward) and 5′-AAATGTGAGCAATTCTGCTT-3′ (reverse); *N-cadherin*, 5′-GCCCCTCAAGTGTTACCTCAA-3′ (forward) and 5′-AGCCGAGTGATGGTCCAATTT-3′ (reverse); and *GAPDH*, 5′-AATCCCATCACCATCTTCCA-3′ (forward) and 5′-TGGACTCCACGACGTACTCA-3′ (reverse) [35,36]. The relative fold gene expression for each sample was calculated using the 2^−^^ΔΔCT^ method.

### 4.11. Statistical Analysis

All experimental data were presented as the mean ± SD of at least three experiments. Statistical analyses were performed using Student’s *t*-test in Microsoft Excel (Microsoft Corporation, Redmond, WA, USA), with *p* < 0.05 indicating statistical significance.

## Figures and Tables

**Figure 1 ijms-22-08167-f001:**
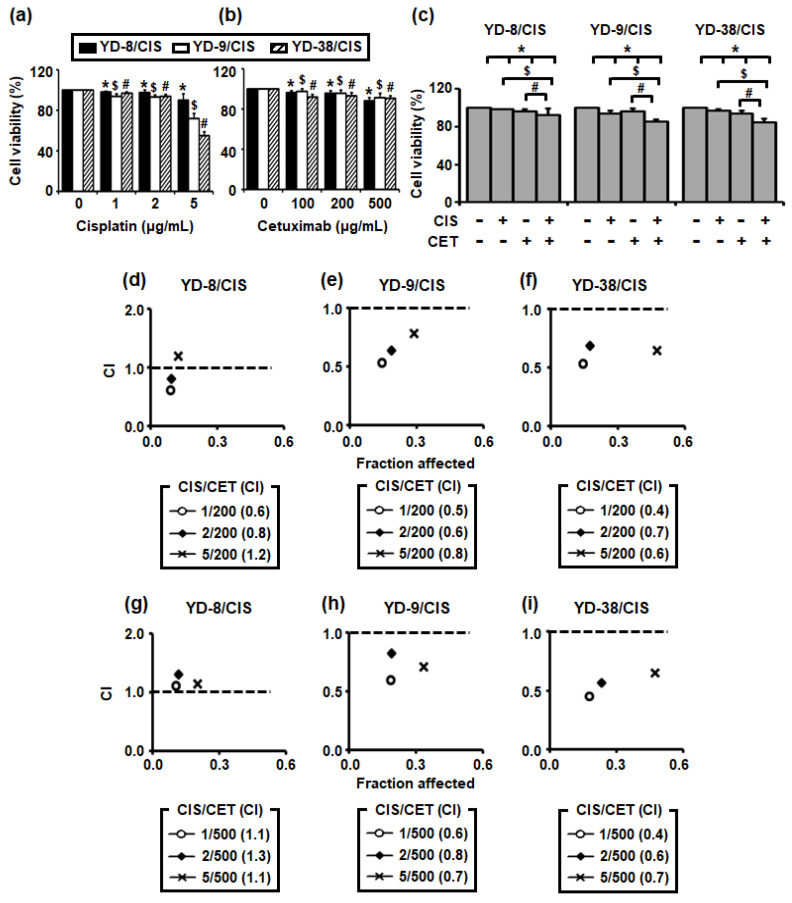
Cisplatin and cetuximab co-treatment had a significant synergistic effect in oral squamous cell carcinoma (OSCC) cancer cell lines. (**a**–**c**) Cells were treated with cisplatin (CIS) (0–5 μg/mL) and/or cetuximab (CET) (0–500 μg/mL) for 72 h. After incubation, combined treatment promoted greater inhibition of cell growth in these cancer cell lines compared to cisplatin (1 μg/mL) or cetuximab (200 μg/mL) treatment alone. The effects of cisplatin with or without cetuximab on the viability of YD-8/CIS, YD-9/CIS, and YD-38/CIS cells were determined using the MTT assay (mean ± standard deviation (SD); *n* = 6). (**a**,**b**) * *p* < 0.05 versus non-treated group in YD-8/CIS cells, $ *p <* 0.05 versus non-treated group in YD-9/CIS cells, and # *p* < 0.05 versus non-treated group in YD-38/CIS cells. (**c**) * *p* < 0.05 versus non-treated group, $ *p <* 0.05 versus only cisplatin-treated group, and # *p* < 0.05 versus only cetuximab-treated group. (**d**–**i**) Synergistic effects were noted in most combinations, except for a few cases in YD-8/CIS cells. The fraction affected versus CI plot (CIS/CET) was determined using the Chou–Talalay and CompuSyn software.

**Figure 2 ijms-22-08167-f002:**
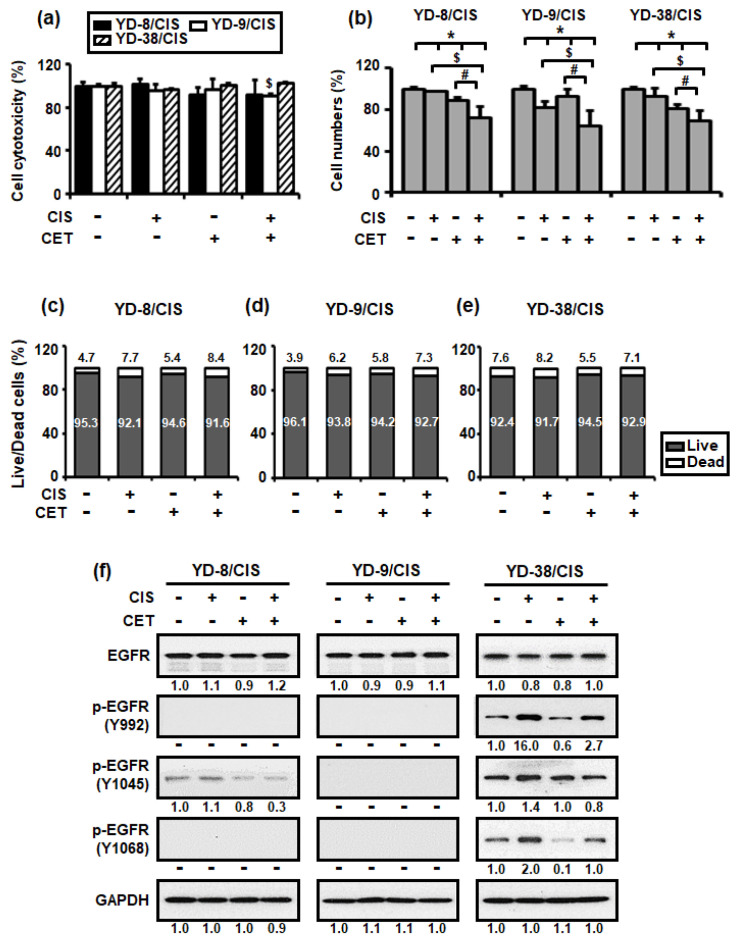
Combined treatment with cisplatin (CIS) (1 μg/mL) and cetuximab (CET) (200 μg/mL) do not induce cell cytotoxicity after 15 h but reduced cell growth after 72 h in cisplatin-resistant OSCC cells. (**a**) Cell cytotoxicity was detected using the LDH assay (mean ± SD; *n* = 6). * *p* < 0.05 versus non-treated group in YD-8/CIS cells, $ *p <* 0.05 versus non-treated group in YD-9/CIS cells, and # *p* < 0.05 versus non-treated group in YD-38/CIS cells. (**b**) The number of cells was measured using the trypan blue assay (mean ± SD; *n* = 6). * *p* < 0.05 versus non-treated group, $ *p <* 0.05 versus only cisplatin-treated group, and # *p* < 0.05 versus only cetuximab-treated group. (**c**–**e**) Live/dead cell ratio was calculated based on the result of (**b**). (**f**) The effects of cetuximab were confirmed using the expression levels of EGFR signaling pathway proteins in cell lysates from all three cell lines treated with cisplatin and/or cetuximab for 72 h. The levels of the indicated proteins (epidermal growth factor receptor (EGFR), phosphor-EGFR (p-EGFR) (Y992, Y1045, Y1068), and glyceraldehyde-3-phosphate dehydrogenase (GAPDH)) were determined using the Western blot assay. Western blot results were quantified as values calculated using the Image J software. The numbers represent the ratio of the optical density of cisplatin and/or cetuximab-treated to non-treated cells normalized by GAPDH.

**Figure 3 ijms-22-08167-f003:**
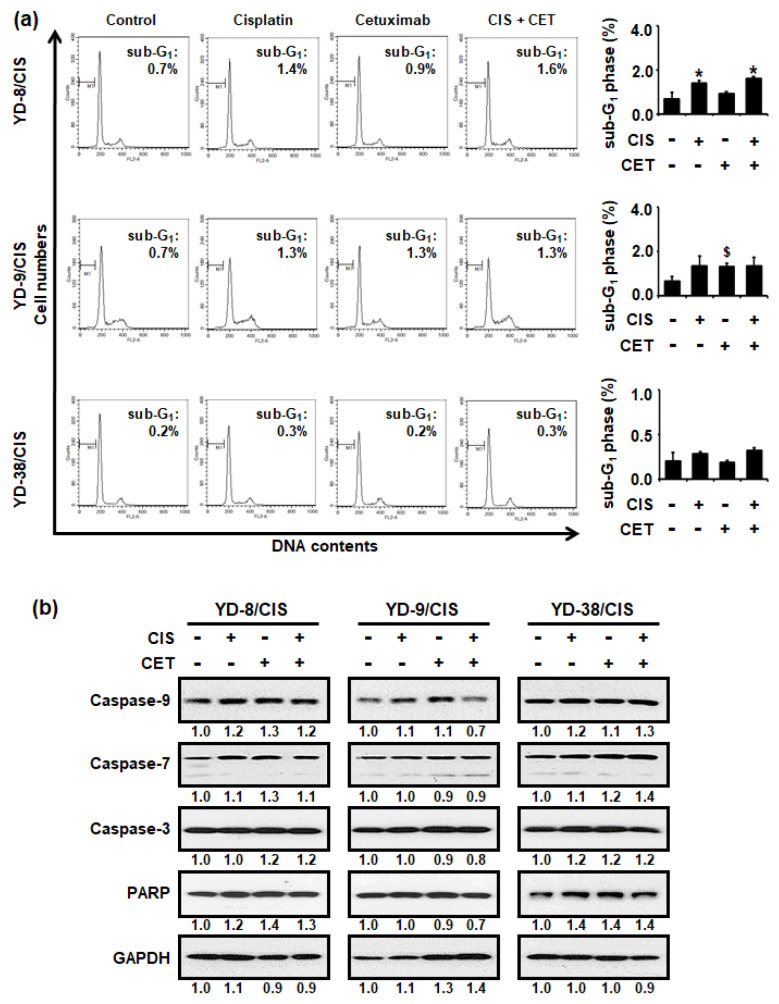
Cisplatin (CIS) (1 μg/mL) and cetuximab (CET) (200 μg/mL) combination for 72 h did not affect apoptosis in cisplatin-resistant OSCC cells. (**a**) Left panel, the cytotoxic effect of cisplatin and/or cetuximab in cisplatin-resistant OSCC cells was measured using PI staining assay. A sub-G1 peak may be present when cells enter apoptosis in the G1 phase. Right panel, graph shows quantitative results for left panel (mean ± SD; *n* = 3). * *p* < 0.05 versus non-treated group in YD-8/CIS cells, # *p <* 0.05 versus non-treated group in YD-9/CIS cells, and $ *p* < 0.05 versus non-treated group in YD-38/CIS cells. (**b**) Levels of apoptosis signaling pathway proteins (Caspase-9, -7, -3, and poly (ADP-ribose) polymerase (PARP)) and glyceraldehyde-3-phosphate dehydrogenase (GAPDH) in cell lysates from all three cell lines treated with cisplatin and/or cetuximab for 72 h were determined using Western blot assay. Western blot results were quantified using Image J software. The numbers represent the ratio of the optical density of cisplatin and/or cetuximab-treated to non-treated cells normalized by GAPDH.

**Figure 4 ijms-22-08167-f004:**
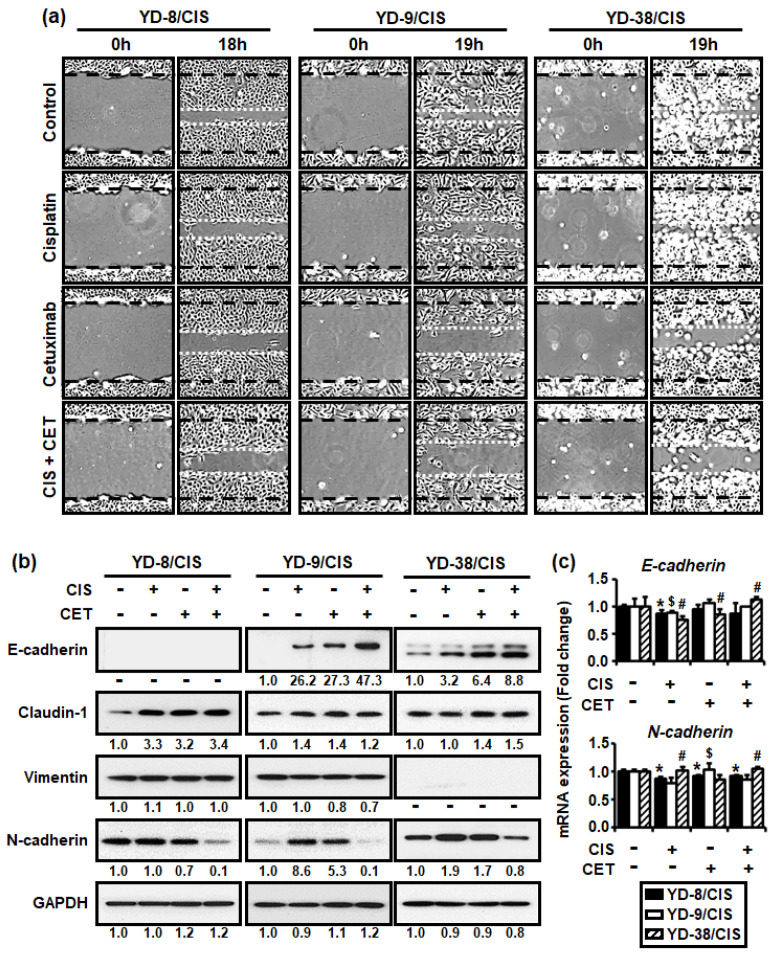
Cisplatin (CIS) (1 μg/mL) and cetuximab (CET) (200 μg/mL) co-treatment promoted greater reduction in cell motility and N-cadherin expression in cisplatin-resistant OSCC cells compared to each alone. However, co-treatment promoted the expression of E-cadherin, which is known to be involved in cell adhesion. (**a**) Cell migration was analyzed using in vitro scratch assay, whereas images were obtained using an Olympus CKX53 inverted microscope (×40). (**b**) The levels of EMT signaling pathway proteins (E-cadherin, claudin-1, vimentin, and N-cadherin) and GAPDH in cell lysates from all three cell lines treated with cisplatin and/or cetuximab for 48 h were determined using the Western blot assay. Western blot results were quantified using Image J software. The numbers represent the ratio of the optical density of cisplatin and/or cetuximab-treated to non-treated cells normalized by GAPDH. (**c**) The expressions of E-cadherin and N-cadherin mRNA from all three cell lines treated with cisplatin and/or cetuximab for 24 h were determined using real-time qRT-PCR (mean ± SD; *n* = 3). * *p* < 0.05 versus non-treated group in YD-8/CIS cells, # *p <* 0.05 versus non-treated group in YD-9/CIS cells, and $ *p* < 0.05 versus non-treated group in YD-38/CIS cells.

## Data Availability

All data are available from the corresponding author upon reasonable request.

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
