# Peer review of "Cisplatin Plus Cetuximab Inhibits Cisplatin-Resistant Human Oral Squamous Cell Carcinoma Cell Migration and Proliferation but Does Not Enhance Apoptosis"

_ijms, 2021, doi:10.3390/ijms22158167_

Round 1

Reviewer 1 Report

This study attempts to demonstrate that cisplatin and cetuximab inhibits proliferation and migration of cisplatin-resistant human oral squamous cell carcinoma cell lines in vitro in a synergistic manner and investigates the effects of this combination therapy on EMT.

Although this study could be of potential interest, the data in current manuscript are rather a patchwork of observations than contributing to a sound story and do not allow to draw overall conclusions. For example, the application of cetuximab reduces the expression of 2 p-EGFR protein in one cell line and of another p-EGFR protein in a second cell line. How does this fit with other data presented? N-cadherin was the only marker to be consistenly downregulated in all 3 cell lines. The other EMT markers investigated are only argulated in one or two out of three cell lines.
Furthermore some of results presented raise some major concerns:

- From the cell viability data presented in Figures 1 a-c, why would one choose the combination of CIS/CET, when 5 µg/mL would be most effective? Furthermore, is the postulated synergistic effect on cell viability statistically significant?

- Similarly, is the synergistic inhibitory effect of the combination treatment on cell numbers statistically different from single treatments (Figure 2 b)?

- Figure 1 g does not show an effective combination index of CIS/CET in YD-8/CIS cells, however, it seems to lead to a decreased percentage of cell numbers (see Figures 2 b, c). What is the explanation of the authors?

- Why was a timepoint of 15 hrs chosen for the LDH-cytotoxicity assay and other measurements were performed after 72 hrs of incubation? Did the authors perform time series experiments?

- Where did the cells disappear in Figures 2 c-e. The sum of live and dead cells should be 100%.

- In Figure 4 f, why is there a double band for the YD-38/CIS cell, which is not visible in the YD-9/CIS cells?

Reviewer 2 Report

In this manuscript, the authors investigated the anticancer efficacy of combination of cisplatin with cetuximab on cisplatin-resistant OSCC. Combination treatment suppresses the cell proliferation, migration and EMT but not apoptosis. These results may relate to the inhibition of EGFR expression. The present results are well presented, but the manuscript needs to provide some information.

  1. Please provide detailed information for the cytotoxicity and cell survival of combination treatment on non-OSCC cells.
  2. Please provide the profile of gene expression of E-cadherin/N-cadherin.

Author Response

Thanks for the comments. We showed the corrections in red according to the suggestions.

Comments and Suggestions for Authors

In this manuscript, the authors
investigated the anticancer efficacy of combination of cisplatin
with cetuximab on cisplatin
-resistant OSCC. Combination treatment suppresses the cell
proliferation, migration and EMT but not apoptosis. These results may relate to the inhibition

of EGFR ex
pression. The present results are well presented, but the manuscript needs to
provide some information.

1.
Please provide detailed information for the cytotoxicity and cell survival of combination
treatment on non
-OSCC cells.
Thank you for your comments. We added the manuscript following your advice.

Recent in vitro studies have reported that a specific EGFR tyrosine kinase inhibitor increased
the therapeutic effects of cisplatin in oral squamous cell carcinoma (OSCC) cells and that
cetuximab increased cisplatin-induced apoptosis through inactivation of EGFR/AKT signaling
pathway in nasopharyngeal carcinoma (NPC), cetuximab plus platinum-based chemotherapy
enhanced overall survival in patients with recurrent or metastatic squamous-cell carcinoma of
the head and neck, and cetuximab improved cisplatin-induces reticulum stress-related
apoptosis in laryngeal squamous cell carcinoma cells by suppressing the expression of
TXNDC5.

2.
Please provide the profile of gene expression of E-cadherin/N-cadherin
Thanks for your comments. We added new data. In new data, we confirmed that the gene
expression of E-cadherin/N-cadherin through the qRT-PCR analysis according to your
comments. The combination of cisplatin and cetuximab had little effect on the expression of
E-cadherin and N-cadherin (Figure 4c).

Round 2

Reviewer 1 Report

Although the authors have tried to address all issues, some major concerns remain:

As long as the authors do not demonstrate statistical significance of the combination treatment in comparison to single treatment, i do not accept the superiority of the combination treatment. There is no reason why a t-test can not be performed with data presented in Figure 1 c. Especially, since the single treatment with 5µg/ml cisplatin has the most pronounced effect on cell viability of all treatments (Figure 1 a).

Similarly, the authors should peform statistics comparing  combination treatment with single treatments and not with controls, to demonstrate superiority in Figure 2b.

The numbers of live/dead cells in Figure should be 100% as in the controls. If you have a cytotoxic effect, than the cells should appear in the dead cell fraction

Author Response

Thanks for the comments. We showed the corrections in red according to the suggestions.
Comments and Suggestions for Authors

Although the authors have tried to address all issues, some major concerns remain:

As long as the authors do not demonstrate statistical significance of the combination treatment

in comparison to single treatment, i do not accept the superiority of the co
mbination treatment.
There is no reason why a t
-test can not be performed with data presented in Figure 1 c.
Thank you for
your comment. We added statistically significant and modified the graph in
Figure
1c based on your advice.
Especially, since the single treatment with 5μ g/ml cisplatin has the most pronounced effect on

cell viability of all treatments (Figure 1 a).

A
s you said, the cell growth was most inhibited at the cisplatin concentration of 5μ g/ml.
However,
our purpose in this study was to find the optimal concentration with a synergistic
eff
ect when cisplatin and cetuximab were co-treated. Therefore, we thought that it was
meaningful to confirm the possibility of
cell migration inhibitory effect with a low
concentration of cisplatin in combination therapy.

Simi
larly, the authors should perform statistics comparing combination treatment with single
treatments and not with controls, to demonstrate superiority in Figure 2b.

Thank you for
your comment. We added statistics and modified the graph in Figure 2b based
on your advice.

The numbers of live/dead cells in Figure should be 100% as in the controls. If you have a

c
ytotoxic effect, than the cells should appear in the dead cell fraction
Thank you for
your comment. We modified the numbers of live/dead cells in Figure 2c-e based
on your advice.

Round 3

Reviewer 1 Report

none